# Unsupervised Semantic Correspondence Using Stable Diffusion

**Eric Hedlin[1], Gopal Sharma[1], Shweta Mahajan[1, 2], Hossam Isack[3], Abhishek Kar[3],**
**Andrea Tagliasacchi[3, 4, 5], Kwang Moo Yi[1]**
[1] University of British Columbia, [2] Vector Institute for AI, [3] Google,
[4] Simon Fraser University, [5] University of Toronto

## Abstract

Text-to-image diffusion models are now capable of generating images that are often indistinguishable from real images. To generate such images, these models must understand the semantics of the objects they are asked to generate. In this work we show that, *without any training*, one can leverage this semantic knowledge within diffusion models to find semantic correspondences – locations in multiple images that have the same semantic meaning. Specifically, given an image, we optimize the *prompt* embeddings of these models for maximum attention on the regions of interest. These optimized embeddings capture semantic information about the location, which can then be transferred to another image. By doing so we obtain results on par with the *strongly supervised* state of the art on the PF-Willow dataset and significantly outperform (20.9% relative for the SPair-71k dataset) any existing weakly or unsupervised method on PF-Willow, CUB-200 and SPair-71k datasets.

## 1 Introduction

Estimating point correspondences between images is a fundamental problem in computer vision, with numerous applications in areas such as image registration [1], object recognition [2], and 3D reconstruction [3]. Work on correspondences can largely be classified into *geometric* and *semantic* correspondence search. Geometric correspondence search aims to find points that correspond to the same physical point of the same object and are typically solved with local feature-based methods [4, 5, 6, 7, 8] and optical flow methods [9, 10]. Semantic correspondence search – the core application of interest in this paper – focuses on points that correspond semantically [11, 12, 13, 14, 15, 16], not necessarily of the same object, but of *similar* objects of the same or related class. For example, given the selected kitten paw in the (source) image of Figure 1, we would like to automatically identify kitten paws in other (target) images. Whether geometric or semantic correspondences, a common

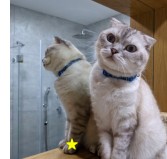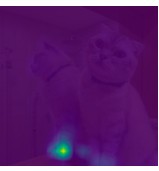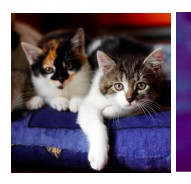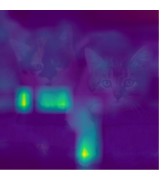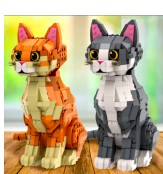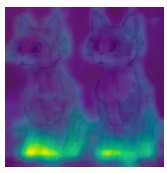

source image    source attention       target image    target attention       target image    target attention

Figure 1: **Teaser** – We optimize for the prompt embedding of the diffusion model that activates attention in the region of the 'paw' for the *source* image, marked in yellow. With this embedding, the attention highlights semantically similar points in various target images, which we then use to find semantic correspondences. This holds even for "out of distribution" target images (LEGO cats).

37th Conference on Neural Information Processing Systems (NeurIPS 2023).

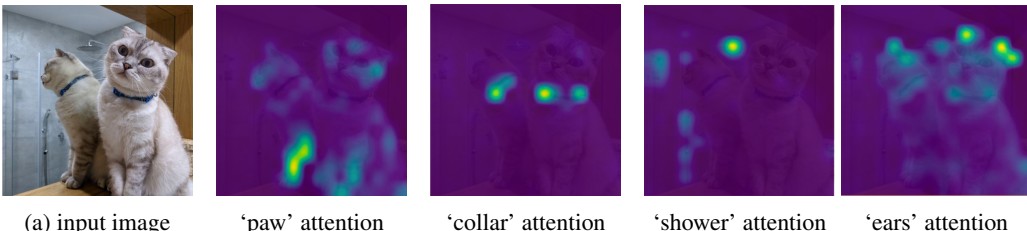

| (a) input image | 'paw' attention | 'collar' attention | 'shower' attention | 'ears' attention |

Figure 2: **Semantic knowledge in diffusion models** – Given an input image and text prompts describing parts of the image, the attention maps of diffusion models highlight semantically relevant areas of the image. We visualize the attention maps superimposed atop the input image.

recent trend is to *learn* to solve these problems [17], as in many other areas of computer vision.

While learning-based methods provide superior performance [16, 15, 17, 18] often these methods need to be trained on large supervised datasets [17, 19]. For geometric correspondences, this is relatively straightforward, as one can leverage the vast amount of photos that exist on the web and utilize points that pass sophisticated geometric verification processes [17, 20, 19]. For semantic correspondences, this is more challenging, as one cannot simply collect more photos for higher-quality ground truth–automatic verification of semantic correspondences is difficult, and human labeling effort is required. Thus, research on unsupervised learned semantic correspondence has been trending [21, 22, 23].

In this work, we show that one may not need any ground-truth semantic correspondences between image pairs at all for finding semantic correspondences, or need to rely on generic pre-trained deep features [24, 25] – we can instead simply harness the knowledge within powerful text-to-image models. Our key insight is that, *given that recent diffusion models [26, 27, 28, 29, 30] can generate photo-realistic images from text prompts only, there must be knowledge about semantic correspondences built-in within them*. For example, for a diffusion model to successfully create an image of a human face, it must know that a human face consists of two eyes, one nose, and a mouth, as well as their supposed whereabouts – in other words, it must know the semantics of the scene it is generating. Thus, should one be able to extract this knowledge from these models, trained with billions of text-image pairs [30, 31], one should be able to solve semantic correspondences as a by-product. Note here that these generative models can also be thought of as a form of unsupervised pre-training, akin to self-supervised pre-training methods [32, 25], but with more supervision from the provided text-image relationship.

Hence, inspired by the recent success of prompt-to-prompt [33] for text-based image editing, we build our method by exploiting the *attention maps* of latent diffusion models. These maps attend to different portions of the image as the text prompt is changed. For example, given an image of a cat, if the text prompt is 'paw', the attention map will highlight the paws, while if the text prompt is 'collar', they will highlight the collar; see Figure 2. Given arbitrary input images, these attention maps should respond to the semantics of the prompt. In other words, if one can identify the 'prompt' corresponding to a particular image location, the diffusion model could be used to identify semantically similar image locations in a new, unseen, image. Note that finding actual prompts that correspond to words is a *discrete* problem, hence difficult to solve. However, these prompts do not have to correspond to actual words, as long as they produce attention maps that highlight the queried image location. In other words, the analogy above holds when we operate on the continuous *embedding* space of prompts, representing the core insight over which we build our method.

With this key insight, we propose to find these (localized) embeddings by *optimizing* them so that the attention map within the diffusion models corresponds to points of interest, similarly to how prompts are found for textual inversion [34, 35]. As illustrated in Figure 1, given an (source) image and a (query) location that we wish to find the semantic correspondences of, we first optimize a randomly initialized text embedding to maximize the cross-attention at the query location while keeping the diffusion model fixed (*i.e.* stable diffusion [30]). We then apply the optimized text embedding to another image (*i.e.* target) to find the semantically corresponding location – the pixel attaining the maximum attention map value within the target image.

Beyond our core technical contribution, we introduce a number of important design choices that deal with problems that would arise from its naive implementation. Specifically, (1) as we optimize

on a single image when finding the embeddings, to prevent overfitting we apply random crops; (2) to avoid the instability and randomness of textual inversion [36, 35] we find multiple embeddings starting from random initialization; (3) we utilize attention maps at different layers of the diffusion network to build a multi-scale notion of semantic matching. These collectively allow our method to be on par with strongly-supervised state of the art on the PF-Willow dataset [13] and outperform all weakly- and un-supervised baselines in PF-Willow, CUB-200 [37], and SPair-71k datasets [14] – on the SPair-71k dataset we outperform the closest weakly supervised baseline by 20.9% relative.

We emphasize that our method does not require supervised training that is specific to semantic point correspondence estimation. Instead, we simply utilize an *off-the-shelf* diffusion model, *without* fine-tuning, and are able to achieve state-of-the-art results. To summarize, we make the following contributions:
- we show how to effectively extract semantic correspondences from an off-the-shelf Stable Diffusion [30] model without training any new task-specific neural network, or using any semantic correspondence labels;
- we introduce a set of design choices – random crops, multiple embeddings, multi-scale attention – that are critical to achieving state-of-the-art performance;
- we significantly outperform prior state of the art based on weakly supervised techniques on the SPair-71k [14], PF-Willow [13], and CUB-200 [37] datasets (20.9% relative on SPair-71k) and is on par with the strongly-supervised state of the art on the PF-Willow [13] dataset.

## 2  Related work

We first review work on semantic correspondences and then discuss work focusing on reducing supervision. We also discuss work on utilizing pre-trained diffusion models.

**Semantic correspondences..** Finding correspondences is a long-standing core problem in computer vision, serving as a building block in various tasks, for example, optical flow estimation [38, 10], structure from motion [39, 40, 3, 5], and semantic flow [11]. While a large body of work exists, including those that focus more on finding geometric correspondences that rely on local features [4, 6, 7] and matching them [41, 42, 43, 8] or directly via deep networks [44, 45, 17, 18], here we focus only on semantic correspondences [11, 16, 15, 46], that is, the task of finding corresponding locations in images that are of the same "semantic" – *e.g.*, paws of the cat in Figure 1. For a wider review of this topic, we refer the reader to [47].

Finding semantic correspondences has been of interest lately, as they allow class-specific alignment of data [46, 48], which can then be used to, for example, train generative models [49], or to transfer content between images [11, 50, 51]. To achieve semantic correspondence, as in many other fields in computer vision, the state-of-the-art is to train neural networks [16, 15]. Much research focus was aimed initially at architectural designs that explicitly allow correspondences to be discovered within learned feature maps [10, 45, 52, 16]. For example using pyramid architectures [52], or architectures [16] that utilize both convolutional neural networks [53] and transformers [54]. However, these approaches require large-scale datasets that are either sparsely [14, 13] or densely [55] labeled, limiting the generalization and scalability of these methods without additional human labeling effort.

**Learning semantic correspondences with less supervision..** Thus, reducing the strong supervision requirement has been of research focus. One direction in achieving less supervision is through the use of carefully designed frameworks and loss formulations. For example, Li *et al.* [21] use a probabilistic student-teacher framework to distill knowledge from synthetic data and apply it to unlabeled real image pairs. Kim *et al.* [22] form a semi-supervised framework that uses unsupervised losses formed via augmentation.

Another direction is to view semantic correspondence problem is utilizing pre-trained deep features. For example, Neural Best-Buddies [56] look for mutual nearest neighbor neuron pairs of a pre-trained CNN to discover semantic correspondences. Amir *et al.* [57] investigate the use of deep features from pre-trained Vision Transformers (ViT) [58], specifically DINO-ViT [25]. More recently, in ASIC [23] rough correspondences from these pre-trained networks have been utilized to train a network that maps images into a canonical grid, which can then be used to extract semantic correspondences.

These methods either rely heavily on the generalization capacity of pre-trained neural network representations [56, 57] or require training a new semantic correspondence network [21, 22, 23]. In

this work, we show how one can achieve dramatically improved results over the current state-of-the-art, achieving performance similar to that of strongly-supervised methods, even without training any new neural network by simply using a Stable Diffusion network.

**Utilizing pre-trained diffusion models..** Diffusion models have recently emerged as a powerful class of generative models, attracting significant attention due to their ability to generate high-quality samples [30, 26, 27, 28, 29, 59, 60]. Diffusion models generate high-quality samples, with text-conditioned versions incorporating textual information via cross-attention mechanisms.

Among them, Stable Diffusion [30] is a popular model of choice thanks to its lightweight and open-source nature. While training a latent diffusion model is difficult and resource-consuming, various methods have been suggested to extend its capabilities. These include model personalization [61] through techniques such as Low Rank Adaptation (LoRA) [62] and textual inversion [34], including new conditioning signals via ControlNet [63], or repurposing the model for completely new purposes such as text-to-3D [64] and text-driven image editing [33]. While the applications vary, many of these applications are for generative tasks, that is, creating images and 3D shapes.

Of particular interest to us is the finding from Prompt-to-Prompt [33] that the cross-attention maps within diffusion models can effectively act as a pseudo-segmentation for a given text query – in other words, it contains semantic information. This is further supported by the fact that intermediate features of diffusion models can be used for semantic segmentation [65]. In this work, with these observations, and with the help of textual inversion [34], we show that we can utilize Stable Diffusion [30] for not just generative tasks, but for the task of finding semantic correspondences, a completely different task from what these models are trained for, all without any training by repurposing the attention maps, similarly as in [66] but for an entirely different application and framework.

Concurrently to our work, work utilizing feature representations within Stable Diffusion for correspondences has been proposed [67, 68, 69]. These methods look into how to use the deep features within the Stable Diffusion Network effectively, similar to how VGG19 features [70] are widely used for various tasks. Our method, on the other hand, looks into how we can alter the attention maps within Stable Diffusion to our will, in other words taking into account how these features are supposed to be used within Stable Diffusion – we optimize word embeddings. By doing so we show that one can perform alternative tasks than simple image creation such as the semantic correspondence task we demonstrated. However, this is not the end of what our framework can do. For example, a recent followup to our preprint demonstrated an extension of our method to segmentation [71]

## 3 Method

Our method identifies semantic correspondences between pairs of images by leveraging the attention maps induced by latent diffusion models. Given a pair of images (respectively source and target), and a query point in the source image domain, we seek to compute an attention map that highlights areas in the target image that are in semantic correspondence with the query. While classical diffusion models work on images directly, we employ latent diffusion models [30], which operate on encoded images. Further, rather than being interested in the generated image, we use the attention maps generated as a by-product of the synthesis process. Our process involves two stages; see Figure 3. In the first stage (optimization), we seek to find an embedding that represents the semantics of the query region in the source image by investigating the activation map of the denoising step at time $t$. In the second stage (inference), the embeddings from the source image are kept fixed, and attention maps for a given target image again at time $t$ are computed. The location attaining the highest value of attention in the generated map provides the desired semantic correspondence. We start by reviewing the basics of diffusion models in Section 3.1, detail our two-stage algorithm in Section 3.2, and augmentation techniques to boost its performance in Section 3.3.

### 3.1 Preliminaries

Diffusion models are a class of generative models that approximate the data distribution by denoising a base (Gaussian) distribution [59]. In the forward diffusion process, the input image $\mathbf{I}$ is gradually transformed into Gaussian noise over a series of $T$ timesteps. Then, a sequence of denoising iterations $\epsilon_{\boldsymbol{\theta}}(\mathbf{I}_t, t)$, parameterized by $\boldsymbol{\theta}$, and $t = 1, \ldots, T$ take as input the noisy image $\mathbf{I}_t$ at each timestep and

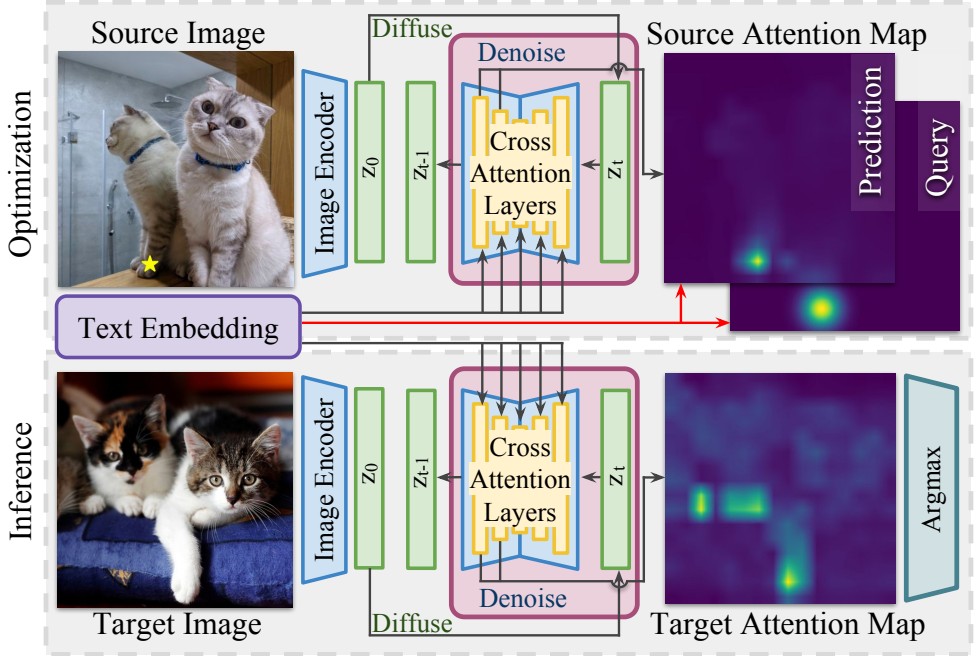

Figure 3: **Method** – (Top) Given a source image and a query point, we *optimize* the embeddings so that the attention map for the denoising step at time $t$ highlights the query location in the source image. (Bottom) During inference, we input the target image and reuse the embeddings for the same denoising step $t$, determining the corresponding point in the target image as the argmax of the attention map. The architecture mapping images to attention maps is a pre-trained Stable Diffusion model [30] which is kept frozen throughout the entire process.

predict the noise added for that iteration $\epsilon$. The diffusion objective is given by:

$$\mathcal{L}_{\text{DM}} = \mathbb{E}_{\mathbf{I}, t, \epsilon \sim \mathcal{N}(0,1)} \left[ \|\epsilon - \epsilon_{\boldsymbol{\theta}}(\mathbf{I}_t, t)\|_2^2 \right]. \tag{1}$$

Rather than directly operating on images $\mathbf{I}$, latent diffusion models (LDM) [30] execute instead on a *latent* representation $\mathbf{z}$, where an encoder maps the image $\mathbf{I}$ into a latent $\mathbf{z}$, and a decoder maps the latent representation back into an image. The model can additionally be made conditional on some text $\mathbf{y}$, by providing an embedding $\mathbf{e} = \tau_{\boldsymbol{\theta}}(\mathbf{y})$ using a text encoder $\tau_{\boldsymbol{\theta}}$ to the denoiser:

$$\mathcal{L}_{\text{LDM}} = \mathbb{E}_{\mathbf{z}, t, \epsilon \sim \mathcal{N}(0,1)} [\|\epsilon - \epsilon_{\boldsymbol{\theta}}(\mathbf{z}_t, t, \mathbf{e})\|_2^2]. \tag{2}$$

The denoiser for a text-conditional LDM is implemented by a transformer architecture [59, 30] involving a combination of self-attention and cross-attention layers.

## 3.2 Optimizing for correspondences

In what follows, we detail how we compute attention masks given an image/embedding pair, how to optimize for an embedding that activates a desired position in the source image, and how to employ this optimized embedding to identify the corresponding point in the target image.

**Attention masks.** Given an image $\mathbf{I}$, let $\mathbf{z}(t)$ be its latent representation within the diffusion model at the $t$-th diffusion step. We first compute query $\mathbf{Q}_l = \Phi_l(\mathbf{z}(t{=}8))^1$, and key $\mathbf{K}_l = \Psi_l(\mathbf{e})$, where $\Phi_l(\cdot)$, and $\Psi_l(\cdot)$ are the $l$-th linear layers of the U-Net [72] that denoises in the latent space. The cross-attention at these layers are then defined as:

$$\mathbf{M}'_l(\mathbf{e}, \mathbf{I}) = \text{CrossAttention}(\mathbf{Q}_l, \mathbf{K}_l) = \text{softmax}\left(\mathbf{Q}_l \mathbf{K}_l^\top / \sqrt{d_l}\right), \tag{3}$$

where the attention map $\mathbf{M}' \in \mathbb{R}^{C \times (h \times w) \times P}$, and $P$, $C$, $h$, and $w$ respectively represent the number of tokens, the number of attention heads in the transformer layer, the height, and width of the image

---

[1]We select the $t{=}8$ diffusion step of a $T{=}50$ steps diffusion model via hyper-parameter tuning.

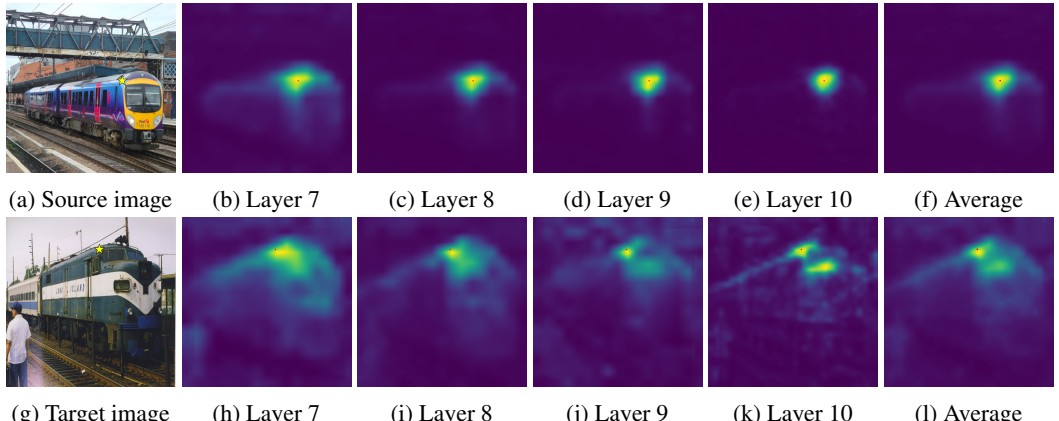

| (a) Source image | (b) Layer 7 | (c) Layer 8 | (d) Layer 9 | (e) Layer 10 | (f) Average |
|---|---|---|---|---|---|
| (g) Target image | (h) Layer 7 | (i) Layer 8 | (j) Layer 9 | (k) Layer 10 | (l) Average |

Figure 4: **Attention response of different layers** – We show example attention maps (b–e) for particular U-Net layers for the optimized embedding that correspond to the location marked with the yellow star on the source image (a) and corresponding average in image (f). We use the same embedding for another (target) image (g) and display its attention map as well (h–k) and their average (l). The ground-truth semantically corresponding region in the target image is marked also with the yellow star. Earlier layers respond more broadly, whereas later layers are more specific. To utilize these different characteristics of each layer we average that of multiple layers into a single attention map.

at the particular layer in the U-Net. Here, $\mathbf{Q} \in \mathbb{R}^{(h \times w) \times d_l}$, and $\mathbf{K} \in \mathbb{R}^{P \times d_l}$ where $d_l$ represents the dimensionality of this layer, and softmax denotes the softmax operation along the P dimension.

As mentioned earlier in Section 1, and observed in [65], different layers of the U-Net exhibit different "level" of semantic knowledge; see Figure 4. Thus, to take advantage of the different characteristics of each layer, we average along both the channel axis and across a subset of U-Net layers $\mathbf{M}'' \in \mathbb{R}^{(h \times w) \times P} = \text{average}_{c=1\dots C, l=7\dots10}(\mathbf{M}'_l)$. Note that the size of the attention maps $\mathbf{M}'_l$ differ according to each layer, hence we utilize bilinear interpolation when averaging. Hence, with $\mathbf{M}(\mathbf{u}; \mathbf{e}, \mathbf{I})$ we denote indexing a pixel $\mathbf{u}$ of the attention map $\mathbf{M}''[1] \in \mathbb{R}^{(h \times w)}$ via bilinear interpolation, where [1] extracts the first of the $P$ available attention maps[2], that is, we use the *first* token of the embedding to represent our query. [3]Examples of these attention maps for embeddings $\mathbf{e}$ derived from natural text are visualized in Figure 2.

**Optimization**. Given a source image $\mathbf{I}_i$ and a query pixel location $\mathbf{u}_i \in [0, 1]^2$, we are interested in finding the corresponding pixel $\mathbf{u}_j$ in the target image $\mathbf{I}_j$. We *emulate* the source attention map for the query as a Gaussian of standard deviation $\sigma$ centered at the query location $\mathbf{u}_i$:

$$\mathbf{M}_s(\mathbf{u}) = \exp\left(-\|\mathbf{u} - \mathbf{u}_i\|_2^2 / 2\sigma^2\right). \tag{4}$$

The Gaussian map represents the *desired* region of focus of the attention mechanism. We then optimize for the (text) embedding $\mathbf{e}$ that reproduces the desired attention map as:

$$\mathbf{e}^* = \arg\min_{\mathbf{e}} \sum_{\mathbf{u}} \|\mathbf{M}(\mathbf{u}; \mathbf{e}, \mathbf{I}_i) - \mathbf{M}_s(\mathbf{u})\|_2^2, \tag{5}$$

**Inference**. With the optimized embedding $\mathbf{e}^*$, we can then identify the target point (the point $\mathbf{u}_j$ in $\mathbf{I}_j$ most semantically similar to $\mathbf{u}_i$ in $\mathbf{I}_i$) by computing the attention map for the target image, and finding the spatial location that maximizes it. We write:

$$\mathbf{u}_j = \arg\max_{\mathbf{u}} \ \mathbf{M}(\mathbf{u}; \mathbf{e}^*, \mathbf{I}_j). \tag{6}$$

---

[2]We do not employ the first or last token as typically these are special termination tokens.

[3]We empirically observed that the choice of which token does not have a significant impact on the final outcome as all other tokens (*i.e.*, $P$ entries of $\mathbf{e}$) all are also optimized regardless due to the softmax that we apply along the $P$ axis – optimization will find the prompts (or more exactly the embeddings) that match the chosen token location.

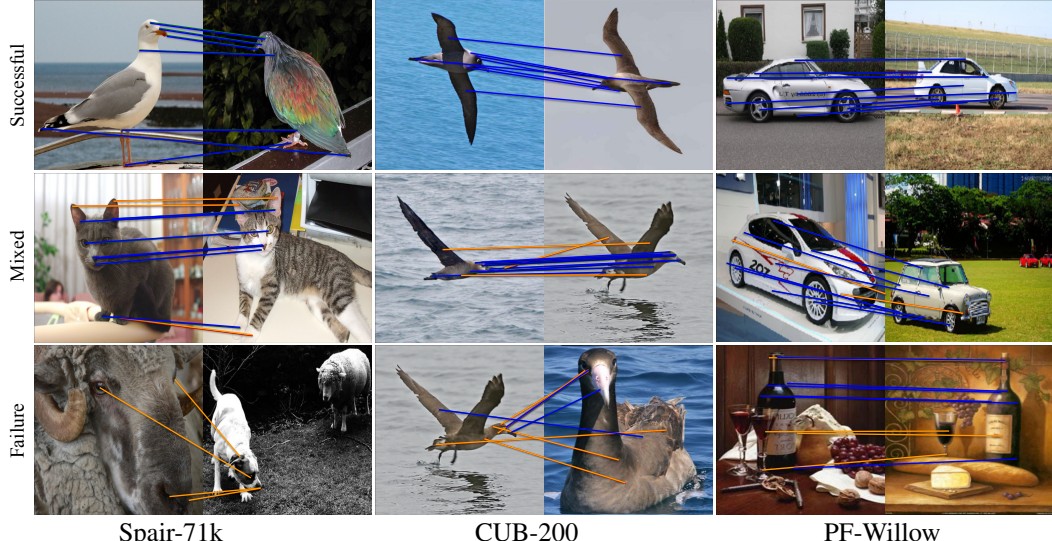

Successful | Mixed | Failure

Spair-71k          CUB-200          PF-Willow

Figure 8: **Qualitative examples** – correspondences estimated from our method are colored in **blue** if correct and in **orange** if wrong according to $PCK_{@0.05}$. (Top) successful cases, (middle) mixed cases, and (bottom) failure cases. Interestingly, even when our estimates disagree with the human annotation (thus shown as orange), they are arguably at plausible locations.

### 3.3 Regularizations

As discussed in Section 1, optimizing text embeddings on a single image makes it susceptible to overfitting. Furthermore, the process of finding embeddings via optimization, textual inversion, has recently been shown to be sensitive to initialization [35, 36]. To overcome these issues, we apply various forms of regularization. Note that while we describe these one at a time, these two augmentations are simultaneously enabled.

**Averaging across crops**. Let $\mathcal{C}_{\mathbf{c}}(\mathbf{I})$ be a cropping operation with parameters $\mathbf{c}$, and $\mathcal{U}_{\mathbf{c}}(\mathbf{I})$ the operation placing the crop *back* to its original location; that is, $\mathcal{C}_{\mathbf{c}}(\mathcal{U}_{\mathbf{c}}(\mathbf{x_c})) = \mathbf{x_c}$ for some crop $\mathbf{x_c}$. For the $\mathbf{c}$ we reduce the image dimensions to 93% (found via hyperparameter sweep) and sample a uniformly random translation within the image – we will denote this sampling as $\mathbf{c} \sim \mathbf{D}$. We augment the optimization in Equation 5 by averaging across cropping augmentations as:

$$\mathbf{e}^* = \arg\min_{\mathbf{e}} \ \mathbb{E}_{\mathbf{c}\sim\mathbf{D}} \ \sum_{\mathbf{u}} \|\mathcal{C}_{\mathbf{c}}(\mathbf{M}(\mathbf{u}; \mathbf{e}, \mathbf{I}_i)) - \mathcal{C}_{\mathbf{c}}(\mathbf{M}_s(\mathbf{u}))\|_2^2, \tag{7}$$

similarly, at inference time, we average the attention masks generated by different crops:

$$\mathbf{u}_j = \arg\max_{\mathbf{u}} \ \mathbb{E}_{\mathbf{c}\sim\mathbf{D}} \ \mathcal{U}_{\mathbf{c}}(\mathbf{M}(\mathbf{u}; \mathbf{e}^*, \mathcal{C}_{\mathbf{c}}(\mathbf{I}_j))). \tag{8}$$

**Averaging across optimization rounds**. We empirically found that doing multiple rounds of optimization is one of the keys to obtaining good performance; see Figure 9. To detail this process, let us abstract the optimization in Equation 7 as $\mathbf{e}^* = \mathcal{O}(\bar{\mathbf{e}}, \mathbf{I}_i)$, where $\bar{\mathbf{e}}$ is its initialization. We then average the attention masks induced by *multiple* optimization runs as:

$$\mathbf{u}_j = \arg\max_{\mathbf{u}} \ \mathbb{E}_{\bar{\mathbf{e}}\sim\mathbf{D}} \ \mathbf{M}(\mathbf{u}; \mathcal{O}(\bar{\mathbf{e}}, \mathbf{I}_i), \mathbf{I}_j). \tag{9}$$

## 4 Results

We evaluate semantic correspondence search on three standard benchmarks: SPair-71k [14] is the largest standard dataset for evaluating semantic correspondences composed of $70,958$ image pairs of 18 different classes. Since we do not perform any training, we only use the $12,234$ correspondences of the test set for evaluation; PF-Willow [13] comprises four classes – wine bottle, duck, motorcycle,

Table 1: **Quantitative results** – We report the Percentage of Correct Keypoints (PCK), where bold numbers are the best results amongst weakly- or un-supervised methods. Our method outperforms all weakly supervised baselines (we use the numbers reported in the literature). Note also that for PF-Willow, our method outperforms even the strongly-supervised state of the art in terms of $PCK_{@0.1}$.

| | | CUB-200 | | PF-Willow | | SPair-71k | |
|---|---|---|---|---|---|---|---|
| | | $PCK_{@0.05}$ | $PCK_{@0.1}$ | $PCK_{@0.05}$ | $PCK_{@0.1}$ | $PCK_{@0.05}$ | $PCK_{@0.1}$ |
| Strong supervision | PWarpC-NC-Net*$_{res101}$ | - | - | 48.0 | 76.2 | 21.5 | 37.1 |
| | CHM | - | - | 52.7 | 79.4 | 27.2 | 46.3 |
| | VAT | - | - | 52.8 | 81.6 | 35.0 | 55.5 |
| | CATs++ | - | - | 56.7 | 81.2 | – | 59.8 |
| Weak supervision | PMD | - | - | 40.3 | 74.7 | – | 26.5 |
| | PSCNet-SE | - | - | 42.6 | 75.1 | – | 27.0 |
| | VGG+MLS | 18.3 | 25.8 | 41.2 | 63.2 | – | 27.4 |
| | DINO+MLS | 52.0 | 67.0 | 45.0 | 66.5 | – | 31.1 |
| | PWarpC-NC-Net$_{res101}$ | – | – | 45.0 | 75.9 | 18.2 | 35.3 |
| | ASIC | 57.9 | 75.9 | **53.0** | 76.3 | – | 36.9 |
| Unsupervised | DINO+NN | 52.8 | 68.3 | 40.1 | 60.1 | – | 33.3 |
| | Our method | **61.6** | **77.5** | **53.0** | **84.3** | **28.9** | **45.4** |

and car – with $900$ correspondences in the test set; `CUB-200` [37] includes 200 different classes of birds. Following ASIC [46] we select the first three classes, yielding a total of $1,248$ correspondences in the test set.

**Metrics**. We measure the performance of each method via the standard metric [13, 14, 23] of Percentage of Correct Keypoints (PCK) under selected thresholds – we report how many semantic correspondence estimates are within the error threshold from human-annotation. Following the standard protocol [13, 14, 23], for SPair-71k and PF-Willow thresholds are determined with respect to the bounding boxes of the object of interest, whereas for CUB-200, the thresholds are set relative to the overall image dimensions. We report results for thresholds $0.05$ and $0.1$, as annotations themselves are not pixel-perfect due to the nature of semantic correspondences – *i.e.*, there is no *exact* geometric correspondence.

**Implementation details**. We use Stable Diffusion version 1.4 [30]. In our ablations, we find that more augmentation help, but with the caveat of (linearly) requiring more computation. Hence, on CUB-200 and PF-Willow datasets we use 10 optimization rounds for the embeddings and 30 random crops for the inference. For the larger SPair-71k dataset we use less – $5$ embeddings and 20 crops. We choose our hyperparameters based on the *validation* subset of SPair-71k and $PCK_{@0.05}$ via a fully-randomized search and applied them to all datasets. We use the Adam [73] optimizer to find the prompts. For detailed hyperparameter setup see Supplementary Material.

**Qualitative highlights**. We show qualitative examples of the semantic correspondences estimated by our method in Figure 8. Interestingly, our method, even when it disagrees with the human annotation, provides results that can arguably be interpreted as plausible. For example, as shown in the wine bottle example at the bottom-right of Figure 8, source points occluded by the wine glasses are correctly mapped to another wine glass in the target image, which disagrees with the human annotator's label which points to the wine bottle.

**Quantitative results**. We report the performance of our method in Table 1. We compare our method against weakly-supervised baselines [21, 52, 46], as well as baselines from ASIC [46] that are based on general-purpose deep features – using VGG [24] features with Moving Least Squares (MLS)[56] and DINO [25] features with Moving Least Squares (MLS) or Nearest Neighbor (NN) matching.

Our method outperforms all compared weakly supervised baselines. Note that the performance gap in terms of average $PCK_{@0.1}$ compared to the second best method, ASIC [46], is large – 20.9% relative. Note also that in the case of the PF-Willow dataset, our method is on par with the current strongly supervised state-of-the-art. Even in the case of the SPair-71k dataset, our results are comparable to a very recent method VAT [15] besides the few problematic classes – bottle, bus, plant, train, tv. These problematic classes are typically those that exhibit strong symmetry, which we do not explicitly deal with. For detailed per-class results, see our Supplementary Material. Considering that our method is

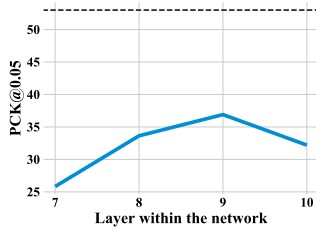 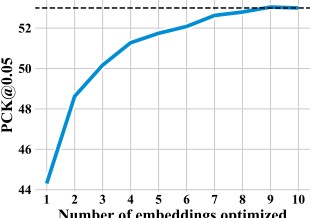 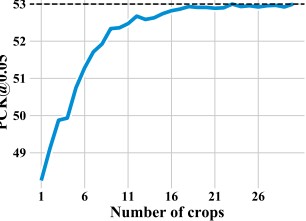

(a) Individual layer performance    (b) # embeddings vs performance    (c) #crops vs performance

Figure 9: **Ablations** – We ablate performance as measured by the $PCK_{@0.05}$ metric on the PF-Willow [13] dataset. (a) Using multiple layers; (b) Using optimization augmentations; (c) Using crop augmentations. Dashed-line denotes the performance of our full method. Note that in (a) individual layer performance is significantly worse, showing that the information within layers is complimentary. In (b) and (c) using more embeddings and crops leads to improved performance.

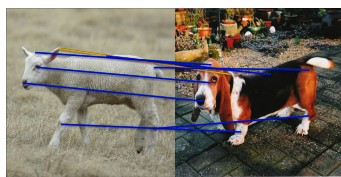 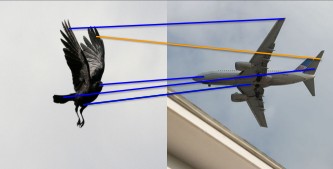 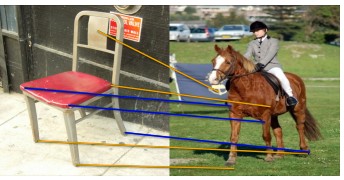

Figure 10: **Semantic correspondence between different classes** – We show examples of applying our method to pairs of image from different classes. We manually mark those that seem arguably correct with **blue**. Our method in many cases generalizes well.

fully unsupervised when it comes to the task of semantic correspondences, these results are quite surprising.

**Ablations**. To ablate our model, we use the PF-Willow [13] dataset and report $PCK_{@0.05}$. In Figure 9a, we show that using individual layers leads to significantly worse results than using *multiple* layers together; this demonstrates the importance of capturing semantic knowledge across multiple receptive fields. In Figure 9b, we show that using multiple optimized embeddings significantly boosts performance, and in Figure 9c, we see how using more crops further leads to improvements during inference. Finally, besides crops during inference, if we disable random crops during embedding optimization $PCK_{@0.05}$ drops to $45.5$ (vs. $53.0$ in our full model).

**Beyond semantic correspondences of the same class.**. We further experiment with correspondences across different classes – *e.g.*, correlating different animals. As shown in Figure 10 in many cases our method provides correspondences that are arguably correct even for objects that are of different classes. This includes being from similar classes (sheep and dog) to more different classes (bird and airplane), and to wildly different classes (chair and horse).

## 5    Conclusions

We have demonstrated the remarkable potential of leveraging diffusion models, specifically Stable Diffusion [30], for the task of semantic correspondence estimation. We have shown that by simply optimizing and finding the embeddings for a location of interest, one can find its semantically corresponding locations in other images, although the diffusion model was never trained for this task. We further introduced design choices that significantly impact the accuracy of the estimated correspondences. Ultimately, our approach significantly outperforms existing weakly supervised methods on SPair-71k [14], PF-Willow [13], and CUB-200 datasets [37] (20.9% relative for SPair-71k), and even achieves performance on par with strongly supervised methods in PF-Willow dataset [13].

**Limitations and future work**. This study highlights the emergent properties of training large models on vast amounts of data, revealing that a model primarily designed for image generation can also be utilized for other tasks, specifically semantic correspondences. The success of our method

underscores the importance of exploring the hidden capabilities of these large text-to-image generative models and suggests that there may be other novel applications and tasks that can benefit from the vast knowledge encoded within these models. For example, as an immediate application, our method can be used to scale up training of 3D generative models such as FigNeRF [49] with images from the web without human supervision.

We note that our method does have its limitations. Many of the failure modes include dealing with symmetric objects. An explicit way to deal with these cases with more refined techniques to extract correspondences may help solve this problem. It also requires significant compute. On an NVIDIA RTX 3090 GPU, finding a single prompt for a single keypoint takes 30 seconds. We suspect that one could potentially train a semantic correspondence network based on our estimates to achieve both scalability in terms of training data and fast inference time.

## Acknowledgments and Disclosure of Funding

This work was supported by the Natural Sciences and Engineering Research Council of Canada (NSERC) Discovery Grant, NSERC Collaborative Research and Development Grant, Google, Digital Research Alliance of Canada, and by Advanced Research Computing at the University of British Columbia.

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
