# Unsupervised Semantic Correspondence Using Stable Diffusion

## Supplementary Material

In this supplementary material we:

- provide per-category quantitative results on SPair71k dataset;
- provide details of hyper-parameters used in various experiments;
- provide details of the architecture of the neural network;
- and provide additional qualitative results on all datasets.

*For complete reproducibility, we will release the code of our experiments if the manuscript is accepted.*

## A  Detailed results for the `SPair-71k` [14] dataset

We report detailed results for the `SPair-71k` [14] dataset in Table 2. When looking at the per-class performance over the 18 classes in the Spair-71k dataset it can be seen that our method outperforms all weakly supervised methods on 16 out of 18 classes and in many cases (bike, car, motorcycle, plant) we have a substantial margin over these methods. We also greatly reduce the margin to strongly supervised methods and for some classes (bike, chair, motorcycle) we outperform them.

## B  Hyperparameter selection

The hyperparameters are selected by carrying out 50 different runs, where each run involves 50 correspondences randomly subsampled from the validation set for the `SPair-71k` [14] dataset. Due to the limited computation resources available at our disposal, we only used a subset of the validation set for searching the hyperparameters. We note that it is possible that a better set of hyperparameters can be found should one use the complete validation set. The best-performing run was then chosen based on its $\text{PCK}_{@0.1}$ metric. Each run was executed over the same set of 50 correspondences, maintaining consistency across all trials. The variation between these runs lies solely in the hyperparameters used, which were selected as follows:

- **U-Net layers:** Randomly selected within the range of 7 to 15.
- **Learning rate for prompt optimization::** A random value between 0.01 and 5e-4 was chosen for each run.
- **Sigma radius:** Selected randomly in the range of 8 to 32.
- **Noise level:** Randomly chosen within the range $t = 1$ to $t = 10$, where $T = 50$.
- **Number of optimization steps:** Randomly chosen in the range of 100 to 300.
- **Image crop size:** The images were cropped consistently within each run and set randomly in the range 50%-100%.

Table 2: `SPair-71k` [14] **detailed results –** we report detailed results on the SPair71k dataset in terms of $\text{PCK}_{@0.1}$. Bolded numbers are best results amongst weak- or un-supervised methods. Our method outperforms all compared weakly supervised baselines and is comparable to CHM [69], a strongly supervised baseline from 2021. Note that on `PF-Willow` [13] we outperform even strongly supervised ones in terms of $\text{PCK}_{@0.1}$.

| Supervision | Method | Aero | Bike | Bird | Boat | Bottle | Bus | Car | Cat | Chair | Cow | Dog | Horse | Motor | Person | Plant | Sheep | Train | TV | Avg. |
|---|---|---|---|---|---|---|---|---|---|---|---|---|---|---|---|---|---|---|---|---|
| Strong supervision | VAT [15] | 58.8 | 40.0 | 75.3 | 40.1 | 52.1 | 59.7 | 44.2 | 69.1 | 23.3 | 75.1 | 61.9 | 57.1 | 46.4 | 49.1 | 51.8 | 41.8 | 80.9 | 70.1 | 55.5 |
| | CHM [69] | - | - | - | - | - | - | - | - | - | - | - | - | - | - | - | - | - | - | 46.3 |
| | CATs++ [16] | - | - | - | - | - | - | - | - | - | - | - | - | - | - | - | - | - | - | 59.8 |
| Weak supervision (train/test) | PMD [21] | 26.2 | 18.5 | 48.6 | 15.3 | **38.0** | 21.7 | 17.3 | 51.6 | 13.7 | 34.3 | 25.4 | 18.0 | 20.0 | 24.9 | 15.7 | 16.3 | 31.4 | 38.1 | 26.5 |
| | PSCNet-SE [52] | 28.3 | 17.7 | 45.1 | 15.1 | 37.5 | 30.1 | 27.5 | 47.4 | 14.6 | 32.5 | 26.4 | 17.7 | 24.9 | 24.5 | 19.9 | 16.9 | 34.2 | 37.9 | 27.0 |
| Weak supervision (test-time optimization) | VGG+MLS [56] | 29.5 | 22.7 | 61.9 | 26.5 | 20.6 | 25.4 | 14.1 | 23.7 | 14.2 | 27.6 | 30.0 | 29.1 | 24.7 | 27.4 | 19.1 | 19.3 | 24.4 | 22.6 | 27.4 |
| | DINO+MLS [56,70] | 49.7 | 20.9 | 63.9 | 19.1 | 32.5 | 27.6 | 22.4 | 48.9 | 14.0 | 36.9 | 39.0 | 30.1 | 21.7 | 41.1 | 17.1 | 18.1 | 35.9 | 21.4 | 31.1 |
| | DINO+NN [57] | 57.2 | 24.1 | 67.4 | 24.5 | 26.8 | 29.0 | 27.1 | 52.1 | 15.7 | 42.4 | 43.3 | 30.1 | 23.2 | 40.7 | 16.6 | 24.1 | 31.0 | 24.9 | 33.3 |
| | ASIC [46] | **57.9** | 25.2 | 68.1 | 24.7 | 35.4 | 28.4 | 30.9 | 54.8 | 21.6 | 45.0 | 47.2 | 39.9 | 26.2 | **48.8** | 14.5 | 24.5 | 49.0 | 24.6 | 36.9 |
| Unsupervised | Ours | 54.2 | **45.1** | **72.9** | **33.6** | 34.4 | **34.9** | **42.9** | **66.8** | **25.9** | **56.5** | **49.8** | **48.8** | **46.6** | **48.8** | **30.1** | **33.0** | **49.1** | **43.9** | **45.4** |

The hyperparameters selected from this process were as follows:

- **U-Net layers:** 7, 8, 9, and 10 out of 16. These layers correspond to attention maps of dimensions $16 \times 16$ for layers 7 to 9, and $32 \times 32$ for layer 10.
- **Learning rate for prompt optimization:** $2.37 \times 10^{-3}$
- **Sigma radius:** 27.98
- **Noise level:** Added noise of $t = 8$ where $T = 50$
- **Number of optimization steps:** 129
- **Image crop size**: Crop size as a percentage of the original image is $93.17\%$

## C Model architecture

The architecture in Figure 3 is based on the stable diffusion model version 1.4 [30]. This architecture is designed to accept an input image of shape $3 \times 512 \times 512$, which is then passed through an encoder to yield an image of shape $4 \times 64 \times 64$ with channel dimension $C$ as 4. This encoded image is referred to as $\mathbf{z}_0$. In accordance with the Denoising Diffusion Probabilistic Model (DDPM) [59], noise is added to $\mathbf{z}_0$ to generate $\mathbf{z}_t$.

The denoising U-Net architecture for stable diffusion is comprised of a total of 16 layers: 6 layers in the contracting path, 1 layer in the bottleneck, and 9 layers in the expansive path. The progression of the image through these layers, along with the respective dimensions per layer ($d_l$), are as follows:

- Contracting path: $64 \times 64$ ($d_l = 40$) $\rightarrow 64 \times 64$ ($d_l = 40$) $\rightarrow 32 \times 32$ ($d_l = 80$) $\rightarrow 32 \times 32$ ($d_l = 80$) $\rightarrow 16 \times 16$ ($d_l = 160$) $\rightarrow 16 \times 16$ ($d_l = 160$)
- Bottleneck: $8 \times 8$ ($d_l = 160$)
- Expansive path: $16 \times 16$ ($d_l = 160$) $\rightarrow 16 \times 16$ ($d_l = 160$) $\rightarrow 16 \times 16$ ($d_l = 160$) $\rightarrow 32 \times 32$ ($d_l = 80$) $\rightarrow 32 \times 32$ ($d_l = 80$) $\rightarrow 32 \times 32$ ($d_l = 80$) $\rightarrow 64 \times 64$ ($d_l = 40$) $\rightarrow 64 \times 64$ ($d_l = 40$) $\rightarrow 64 \times 64$ ($d_l = 40$)

A typical U-Net layer in text conditioned latent diffusion models [30] is augmented with the cross-attention mechanism for conditioning on the prompts. The queries in this mechanism are the projections of the flattened intermediate representations of the U-Net, and the keys and the values are the projections of the prompt embeddings. The total length of tokens for this model, $P$, is 77 where each token has a dimensionality of 768.

## D Additional results

To provide more in-depth analysis, in Figure 11 we depict the distribution of image pairs according to the ratio of correspondences within the image pair that achieve $\text{PCK}_{@0.1}$. For example, an image pair with all correctly estimated correspondences would fall into the 100% bin, whereas one that has only have of the correspondences correct in 50%. The bins respectively represent the following $\text{PCK}_{@0.1}$ ranges over the dataset: 0.0%-0.0%, 0.0%-9.1%, ... 90.0%-100.0%. For `PF-Willow` [13]

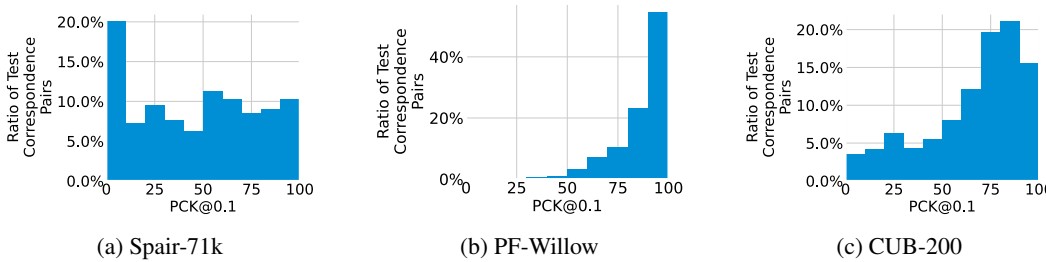

(a) Spair-71k        (b) PF-Willow        (c) CUB-200

Figure 11: **Distribution of image pairs w.r.t correspondence correctness** – We report the distribution of image pairs according to the percent of correspondences within each image that fall under $\text{PCK}_{@0.1}$. For `PF-Willow` [13] and `CUB-200` [37] datasets, majority of image pairs have most correspondences correctly localized, demonstrating more than what the accumulated $\text{PCK}_{@0.1}$ shows. For the harder `SPair-71k` [14] dataset results are spread.

and CUB-200 [37] datasets our approach produces a high $\text{PCK}_{@0.1}$ score for most test image pairs as shown, indicating the effectiveness of our approach. For SPair-71k [14], which is a harder dataset, the results are more evenly spread.

For each of the bin ranges for each dataset, we visualize representative image pairs in Figure 12, Figure 13, and Figure 14. Note that in many cases, incorrectly identified correspondences appear to still align with semantically consistent points on the target object – they simply disagree with the annotated labels of the datasets.

Typical correct and incorrect examples of attention maps for each dataset can be seen in Figure 15, Figure 16, Figure 16, Figure 17, Figure 18, Figure 19, and Figure 20. Correspondences are visualized as lines that connect source points on the left of each image pair to estimated points on the right target image.

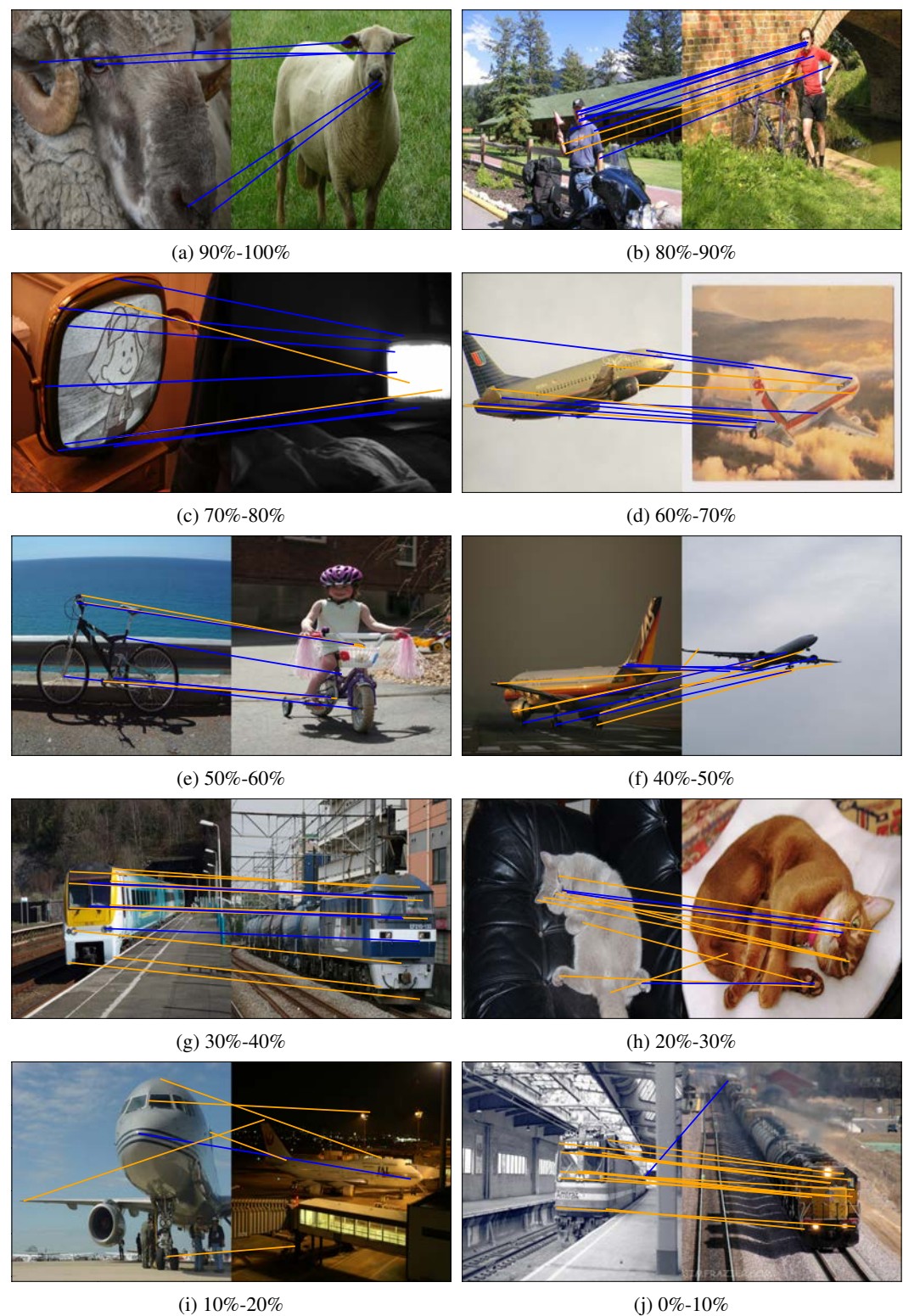

(a) 90%-100%

(b) 80%-90%

(c) 70%-80%

(d) 60%-70%

(e) 50%-60%

(f) 40%-50%

(g) 30%-40%

(h) 20%-30%

(i) 10%-20%

(j) 0%-10%

Figure 12: **Examples for the** SPair-71k **[14] dataset** – typical image pairs for each bin in Figure 11. Correct correspondences are indicated in **blue**, while incorrect ones are depicted in **orange**.

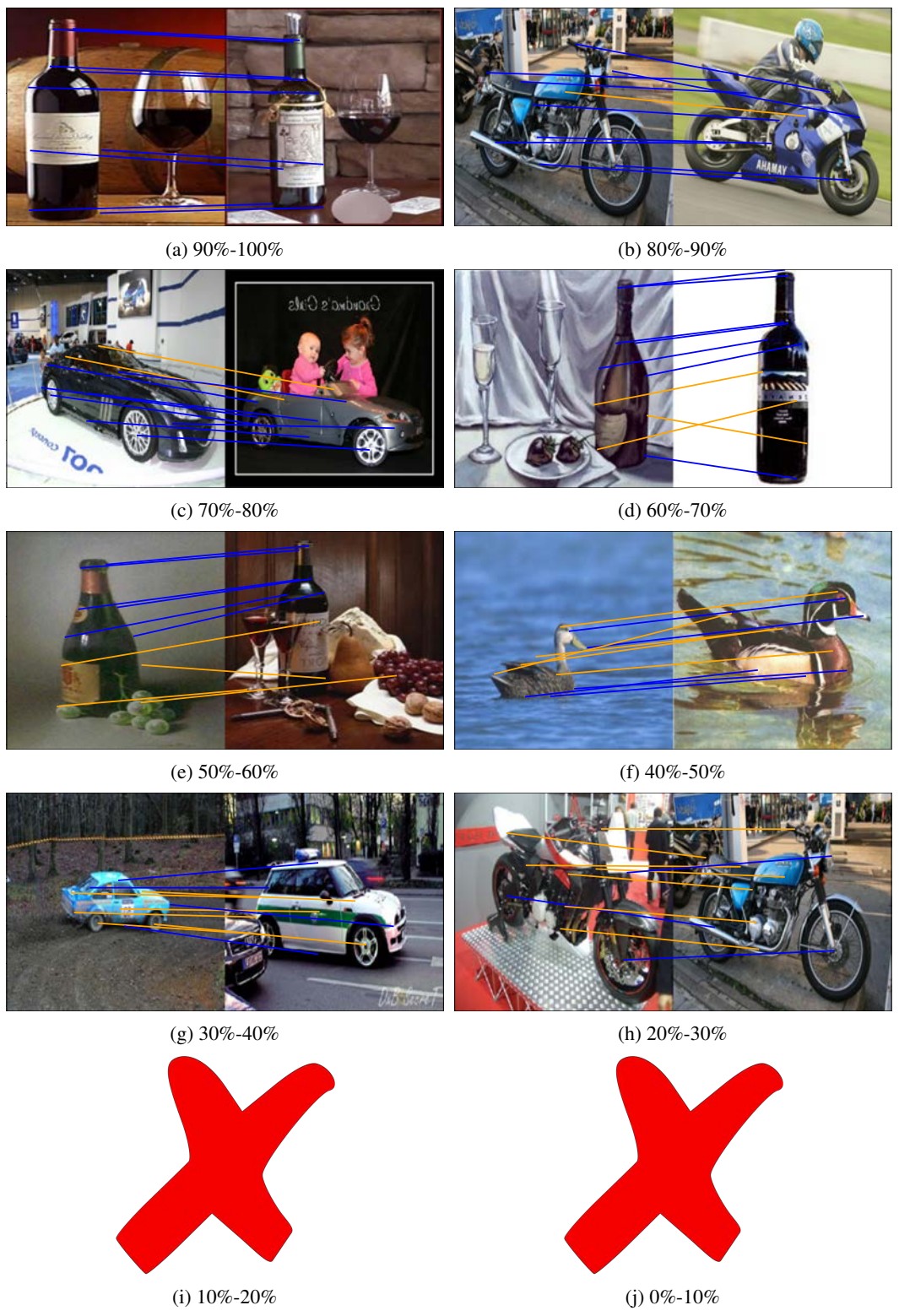

(a) 90%-100%

(b) 80%-90%

(c) 70%-80%

(d) 60%-70%

(e) 50%-60%

(f) 40%-50%

(g) 30%-40%

(h) 20%-30%

(i) 10%-20%

(j) 0%-10%

Figure 13: **Examples for the** `PF-Willow` [13] **dataset** – typical image pairs for each bin in Figure 11. There are no correspondences with accuracies in the range $[0, 30)\%$. Correct correspondences are indicated in **blue**, while incorrect ones are depicted in **orange**.

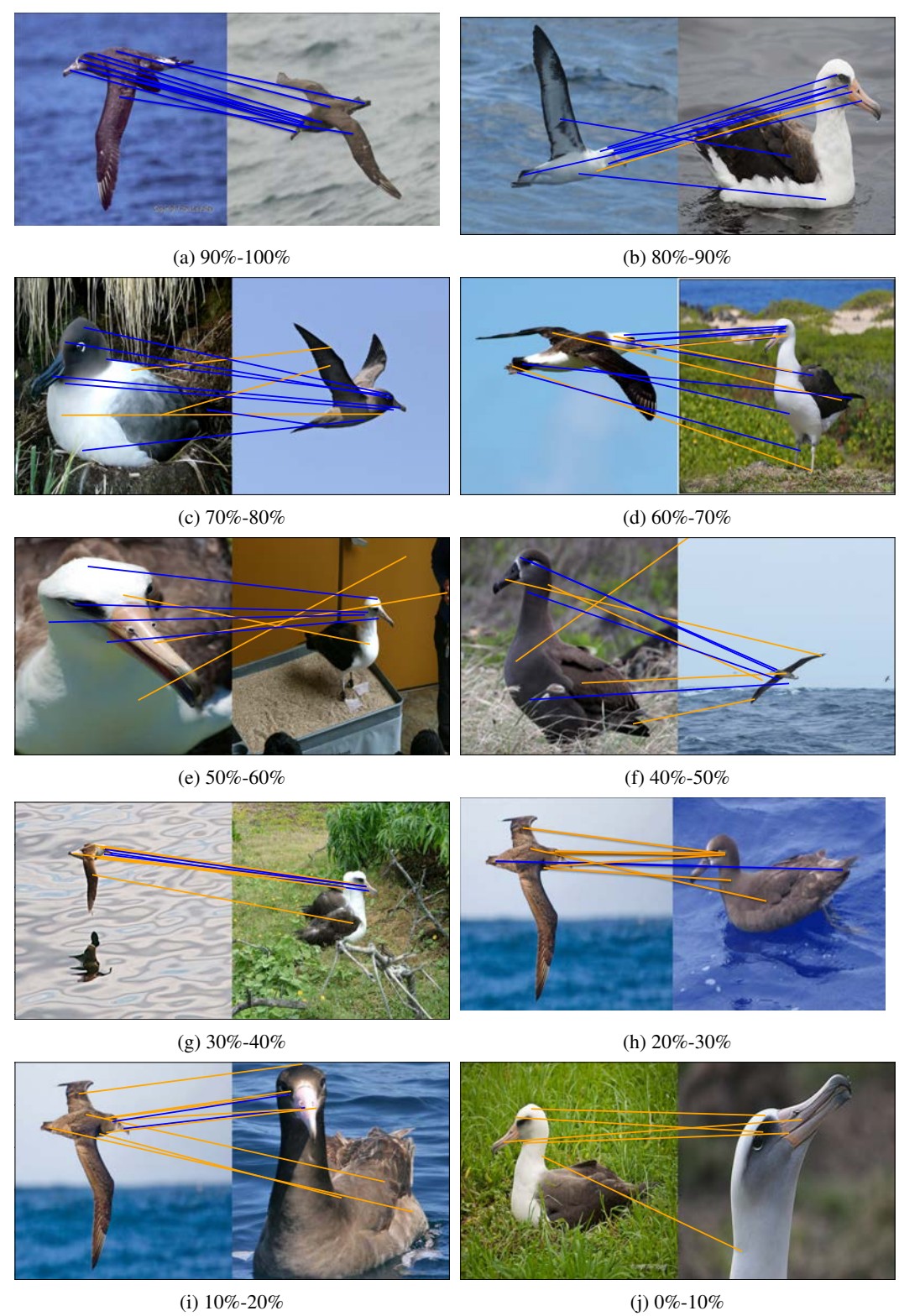

Figure 14: **Examples for the** CUB-200 [37] **dataset** – typical image pairs for each bin in Figure 11. Correct correspondences are indicated in **blue**, while incorrect ones are depicted in **orange**.

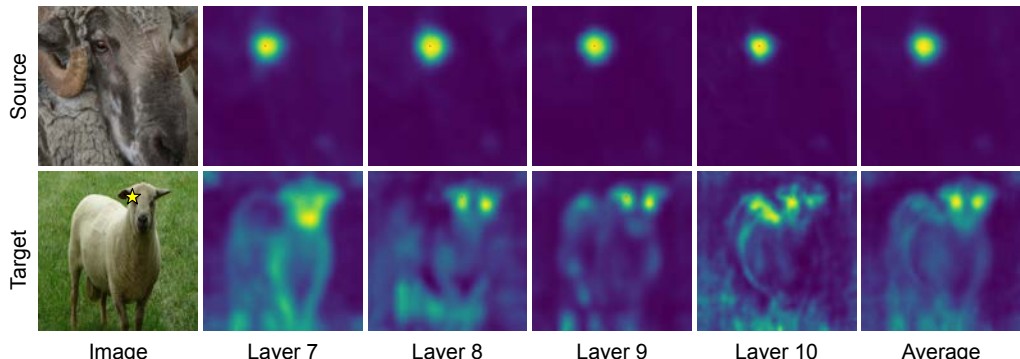

Figure 15: **Correct attention map example for** `SPair-71k` [14] – The model attends to both eyes in the target image, yet it demonstrates a slight preference towards the correct eye. Ground-truth correspondences are marked as yellow star.

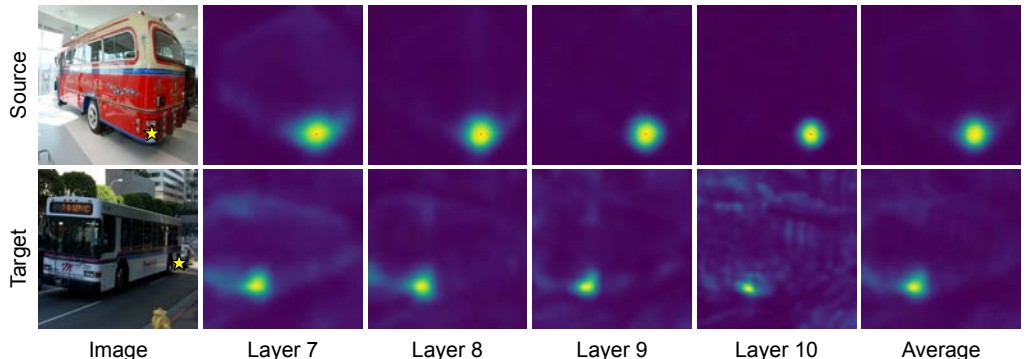

Figure 16: **Incorrect attention map example for** `SPair-71k` [14] – The attention map appears to erroneously concentrate on the near corner of the bus, instead of the front left corner, which is the actual intended correspondence due to symmetry. Ground-truth correspondences are marked as yellow star.

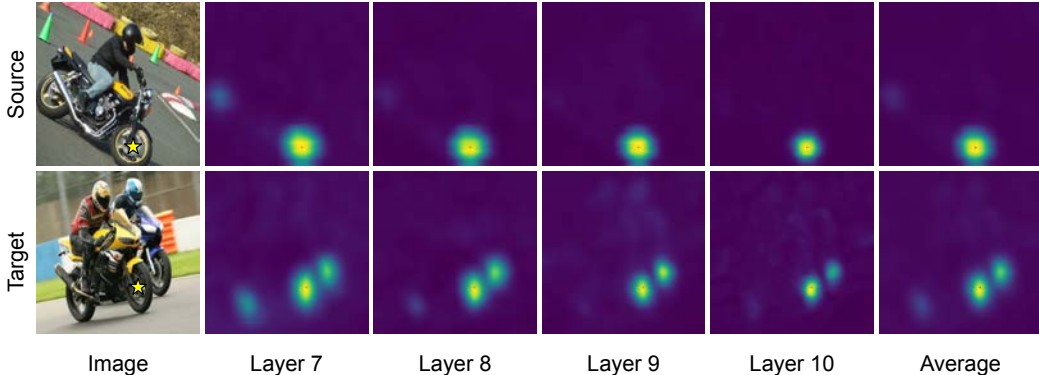

Figure 17: **Correct attention map example for** `PF-Willow` [13] – There are two motorcycles in the target image and attends to the tires of both but still has a preference for the correct correspondence. Ground-truth correspondences are marked as yellow star.

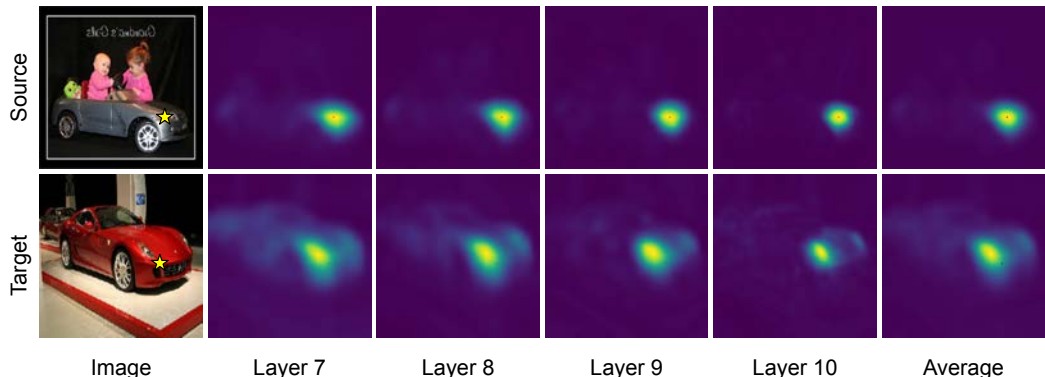

Figure 18: **Incorrect attention map example for** PF-Willow [13] – The attention map in the target image attends to the headlight, which is arguably also correct, as opposed to the corner of the car, which was the intended correspondence by the human labeler. Ground-truth correspondences are marked as yellow star.

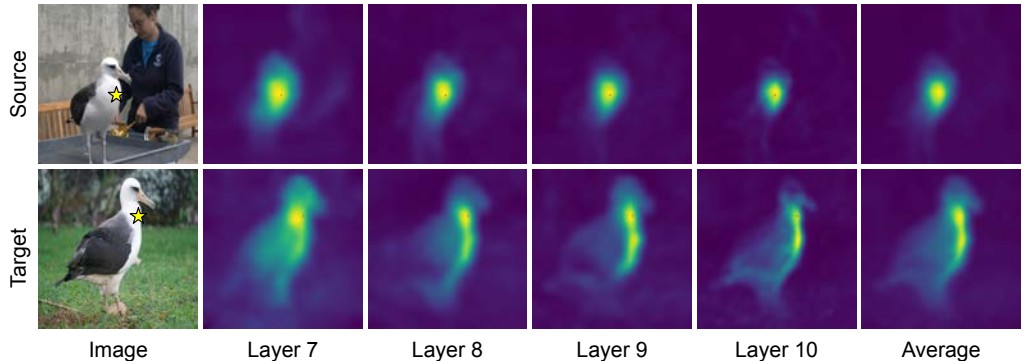

Figure 19: **Correct attention map example for** CUB-200 [37] – The attention map primarily focuses on a line along the bird's front side. Although this shows some uncertainty regarding the precise position of the correspondence, the model nonetheless successfully identifies it. Ground-truth correspondences are marked as yellow star.

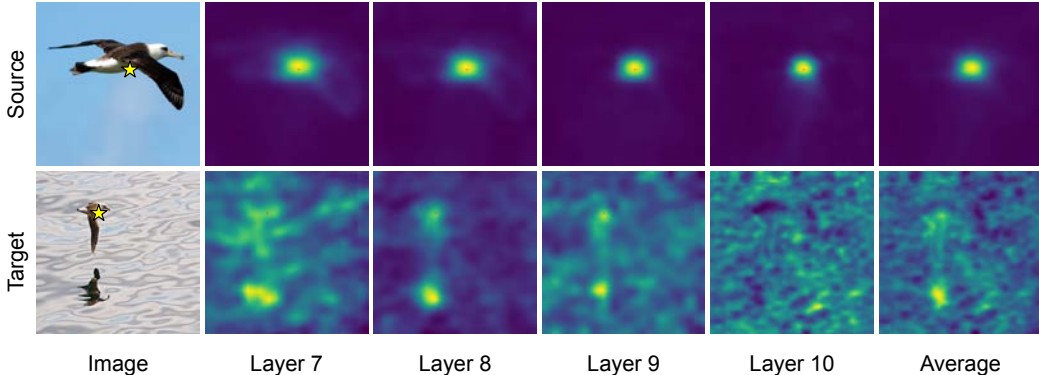

Figure 20: **Incorrect attention map example for** CUB-200 [37] – The attention map for the target image seems to be attending more to the reflection of the bird as opposed to the bird itself. Ground-truth correspondences are marked as yellow star.