# OpenReview forum: "Unsupervised Semantic Correspondence Using Stable Diffusion"
_NeurIPS.cc/2023/Conference — NeurIPS 2023 poster_

### Official Review · Reviewer_DBqD · 2023-06-13

**Soundness:** 2 fair
**Presentation:** 3 good
**Contribution:** 3 good
**Rating:** 6
**Confidence:** 4

**Summary:**

The authors propose an approach to establish semantic correspondences, ie correspondences between different instances of the same object, using a pre-trained stable diffusion models, and without any task-specific finetuning. In particular, they leverage the fact that intermediate attention layers of unet diffusion model respond to the semantic of the text prompt. For a source image and a particular query pixel location for which a match should be computed, they optimize the prompt embedding that leads to a cross-attention map with a high activation at the query location. Once the prompt is recovered, they can apply the reverse attention process on a target image conditioned on the optimized prompt. The intermediate cross-attention layers of the diffusion model highlight the matching location. They also introduce technical ‘tricks’ to make it work, for example averaging over multiple crops of an image or starting from different random embeddings. This process must be repeated for each query pixel location.

**Strengths:**

- The insight that stable diffusion models inherently learn semantic correspondences without task-specific training is interesting. How to extract and use such knowledge has not been explored in the past.

- The results are convincing - the approach obtains near state-of-the-art in semantic matching without any task-specific knowledge or funetuning.

- The paper reads well overall


**Weaknesses:**

1) Missing comparisons and citations in results: PWarpC [Truong et al. CVPR 2022]:
- the paper only includes comparisons to two weakly-supervised semantic matching approaches, with relatively weak results, which allows the authors to claim that their approach is 20% better than the state-of-the-art in weakly-supervised semantic matching.
- The authors did not include results of PWarpC, a weakly-supervised approach for semantic matching presented at CVPR 2022. The results of PWarpC significantly outperform the baseline results of this paper for SPair-71K (38.0 on SPair-71K for PWarpC-NC-Net), making the above claim invalid.
- on PF-Willow: to double check since the two papers use different metrics but PWarpC (weakly-supervised) outperforms CATs, which obtains better metrics than the proposed approach. It might therefore outperform the proposed approach.
- please include this comparisons and revise the claims/analysis accordingly

2) while the paper presents an interesting analysis, the proposed approach is currently very impractical and almost unusable, since as the authors say, it takes 30s to generate a single match. I believe it would be interesting to have even a small experiment training a semantic matching model based on pseudo ground-truth matches obtained from the diffusion model.




**Questions:**

In addition to the weakness above:

A) What is the resolution of the extracted feature/attention maps and at what resolution is the matching done? for example are the features interpolated to higer resolution before computing the cross-attention weights?
Is a similar resolution used in competitor works like DINO?
This is important because the resolution of the feature maps is of crucial importance for obtaining fine-grained/accurate correspondences. It has been the limiting factor of many previous approaches that needed to compute all-to-all correspondences, because of memory constraint. Just using a higher resolution feature map for the cross-attention would lead to an improvement in results, not necessarily related to the features themselves.


B) L.230 the authors mention it takes ‘30s’ to optimize the prompt, does it include only finding the prompt? or finding the match? A real run-time analysis of time versus performance for different settings would be interesting here (depending on how many crops are used, how many initializations ect).

C) clarification needed:
C.1) L.178: M′ has dimension C×(h×w)×P, where is the channel coming from? According to equation, M’ is computed with the standard attention mechanism, which is just a matrix multiplication followed by a softmax. As a result, the dimensions should be (hxw)xP.

C.2) L.185, 187, it would be helpful to specify the shape of M′′ and M, to better follow the different steps.
What is the dimension of P for the learnt embedding?

C.3) L213, is it equation 5 or equation 7? The authors say they average the optimization over multiple crops and over multiple initialization of the embeddings. Is this happening at the same time, i.e. for different initialization, the optimization is done over multiple crops? If so, then it should be equation (7).

C.4) in Tab1, what is meant by ‘weakly-supervised (test-time optimization)’? DINO with NN is for me unsupervised since it only requires single images for the contrastive learning part.


some typos:
L.207 hyperparemter
L.53-54 Note that while finding actual prompt thatcorrespond to words is a discrete problem, hence difficult to solve. -> remove while


**Limitations:**

they have adequately addressed the limitations an negative impact of their work.

---

> ### Author Rebuttal · Authors · 2023-08-10
>
> We thank the reviewer for the constructive feedback and for recognizing that our approach provides new insights for the estimation of semantic correspondences with stable diffusion. We are glad that the reviewer finds our results convincing. We address the reviewer's concerns below:
>
> ### **W-1: Missing comparison with PWarpC**
>
> Thank you for pointing out this baseline. Below is the updated Table with PWarpC. As there are many variants, we have chosen the one that performs the best. Here we provide the updated table. The complete updated table can be found in **Table 1** in the included rebuttal PDF above.
>
> |  | CUB-200 |  | PF-Willow |  | SPair-71k |  |
> | --- | --- | --- | --- | --- | --- | --- |
> |  | PCK@0.05 | PCK@0.10 | PCK@0.05 | PCK@0.10 | PCK@0.05 | PCK@0.10 |
> | Strong supervision |  |  |  |  |  |  |
> | PWarpC-NC-Net* res101 [1] | -- | -- | 48 | 76.2 | 21.5 | 37.1 |
> | Weak supervision |  |  |  |  |  |  |
> | PWarpC-NC-Net res101 [1] | -- | -- | 45 | 75.9 | 18.2 | 35.3 |
> | Unsupervised |  |  |  |  |  |  |
> | DINO+NN [2] | 52.8 | 68.3 | 40.1 | 60.1 | -- | 33.3 |
> | Our method | 61.6 | 77.5 | 53 | 84.3 | 28.9 | 45.4 |
>
> [1] Prune Truong, Martin Danelljan, Fisher Yu, and Luc Van Gool. Probabilistic warp consistency
> for weakly-supervised semantic correspondences. In Conference on Computer Vision and
> Pattern Recognition, 2022.
>
> [2] Shir Amir, Yossi Gandelsman, Shai Bagon, and Tali Dekel. Deep vit features as dense visual
> descriptors. arXiv Preprint, 2021.
>
> ### **W-2 and Q-B: Practicality / Amount of time it takes to optimize the prompt**
>
> It takes 30 seconds to optimize a text embedding; however, inference time is much quicker since it only needs to be run once per crop. For a single crop, it takes around 0.16 seconds, so for the 20 crops we used in our experiments, it takes 3.2 seconds. Inference time can be sped up by reducing the number of crops and Figure 9C of our paper shows the corresponding effect on final performance. This can be helpful for the case where a single optimized keypoint is applied to many images.
>
> We would like to further note that while it takes 30 seconds, there is no network training involved. Other baselines, such as the weakly supervised ones on the table would require training. As an example, the best-performing weakly supervised baseline, ASIC, requires training for around 3 hours per dataset.
>
> Regarding supervising another model with output from our method as pseudo ground truth, due to the limited size of the training dataset, any form of supervised training is shown to have an overfitting effect.
>
> As future research direction for a more practical solution, we will explore optimizing text embeddings for multiple query points jointly that will significantly speed up our approach (also suggested by *reviewer Vgvc*), or methods such as those suggested recently in the textual inversion literature using hypernetworks [3, 4]. We would like to also note that, while early methods have been slow, such as the original NeRF and AlexNet (at the time), follow-ups have significantly sped them up by orders of magnitude, for example as in Instant NGP and MobileNet.
>
> [3]. Ruiz, Nataniel, et al. "HyperDreamBooth: HyperNetworks for Fast Personalization of Text-to-Image Models." *arXiv preprint arXiv:2307.06949* (2023).
>
> [4]. Arar, Moab, et al. "Domain-Agnostic Tuning-Encoder for Fast Personalization of Text-To-Image Models." *arXiv preprint arXiv:2307.06925* (2023).
>
> ### **Q-A: Resolution of the features being used**
>
> We only upsample the attention maps after the cross-attention layer to stay in line with the original stable diffusion model. Cross-attention layers with sizes 16x16 and 32x32 are upsampled to the original image size and then combined.
>
> ### **Q-C: M′ has dimension C×(h×w)×P, where is the channel coming from**
>
> Thank you for pointing this out. We will include the fact in the paper that the cross attention is applied separately over the C channels, leading to the dimension of M′ being C×(h×w)×P.
>
> ### **Q-C.2: shape of M′′ and M and dimension of P**
>
> Thank you for pointing this out. M'' is of shape [(H*W), P]. M is [H, W] and the dimension of P is 77. We will revise to make this clear.
>
> ### **Q-C.3: Is this happening at the same time, i.e. for different initialization, the optimization is done over multiple crops? If so, then it should be equation (7).**
>
> Thank you for pointing out this error. We will correct it. The reviewer is indeed correct that line 213 should be referring to equation (7), not equation (5). This then accurately reflects how the embeddings used at this stage are optimized with different crops.
>
> ### **Q-C.4: DINO with NN is for me unsupervised since it only requires single images for the contrastive learning part.**
>
> Thank you for pointing this out. We will update the table by putting DINO+NN in an unsupervised row. The provided table also reflects this change.

---

> > ### Author Response · Authors · 2023-08-21
> >
> > Thank you for taking into account our rebuttal and raising the score.

---

> ### Comment · Reviewer_DBqD · 2023-08-16
>
> The authors have addressed most of my concerns and those of other reviewers. Despite the significant runtime problem, i think the paper proposes an interesting contributions so will upgrade my score to 6.

---

### Official Review · Reviewer_6281 · 2023-07-01

**Soundness:** 3 good
**Presentation:** 3 good
**Contribution:** 3 good
**Rating:** 7
**Confidence:** 3

**Summary:**

This paper proposes a model that finds semantic correspondence between a pair of images, with a pretrained diffusion model. By optimizing the prompt embeddings, such correspondence can be read out via attention maps in the UNet. Empirical study shows the good performance of this proposed model over exising baselines.

**Strengths:**

1. The proposed method has a simple form but is effective.
2. The presentation is clear and easy to follow.
2. The qualitative results are impressive, especially the results between different classes.
3. Some quantitative results are significantly better than baselines, or comparable to supervised results.

**Weaknesses:**

1. More information should be provided for the optimization set up and implementation, e.g. the optimization algorithm for Eq (5) and Eq (9).
2. The complexity and efficiency are needed for better assessment of the algorithm, especially in comparison to baselines.

**Questions:**

1. I'm wondering how would the trained prompt compare to text embeddings, e.g. how would a prompt optimized from a bird's eye compare to the text embedding of "a bird's left eye"?
2. One textual baseline that doesn't appear in the experiments is the attention map of an actual text prompt, i.e. would the prompt of "a bird's left eye" have its highest attention to the eye's location?

**Limitations:**

The authors has addressed valid limitations of this work.

---

> ### Author Rebuttal · Authors · 2023-08-10
>
> We thank the reviewer for constructive feedback. We appreciate that the reviewer finds our approach effective and our qualitative results across classes impressive. We address the reviewer's concerns below:
>
> ### **W-1: More information about optimization setup and implementation**
>
> We provide detailed information about the implementation in the Supplementary, as it is difficult to fit into the main paper.
>
> We guarantee we will release the code upon acceptance of the manuscript to facilitate reproduction.
>
> ### **W-2: The complexity and efficiency are needed for a better assessment of the algorithm, especially in comparison to baselines.**
>
> We only optimize the text embeddings, hence 59k parameters. No stable diffusion weights are touched. As reported in the submission (Section 5), this results in ~30 seconds per single prompt on a 3090 GPU. Note that while this is the case, compared to baselines, this could result in less compute time if evaluating only a few images -- this is because the weakly supervised baselines require training a deep network (which takes approximately 3 hours per dataset for the best-performing baseline, ASIC), which is not required by our method.
>
> ### **Q-1 and Q-2: How would a prompt optimized from a bird's eye compare to the text embedding of "a bird's left eye"? What does the attention map look like?**
>
> To effectively visualize the optimized embeddings we look into the attention maps for the optimized embedding and the natural prompt.
>
> In the provided rebuttal PDF above, **Figure 2** illustrates an initial image of a bird, while **Figure 3** shows the corresponding optimized attention map focused on the bird's eye. The optimized attention map notably aligns with the Gaussian distribution it was supervised to imitate.
>
> Furthermore, **Figure 5** in the same PDF also visualizes the attention maps for each token corresponding to the sentence "a bird's left eye.” The bird’s eye gets highlighted as we would expect, but not as localized as the optimized embedding.

---

> > ### Comment · Reviewer_6281 · 2023-08-16
> >
> > I thank the authors for the rebuttal. It has addressed most of my concerns and I'd raise the score to 7.

---

> ### Author Response · Authors · 2023-08-21
>
> Thank you for taking into account our rebuttal and raising the score.

---

### Official Review · Reviewer_Vgvc · 2023-07-07

**Soundness:** 2 fair
**Presentation:** 3 good
**Contribution:** 3 good
**Rating:** 5
**Confidence:** 4

**Summary:**

This paper focuses unsupervised semantic correspondence and proposes to leverage the semantic knowledge within fashionable text-to-image diffusion model to accomplish this task. The method optimizes a learnable text prompt to maximize source image attention value of query location. Then, the optimized text prompt is used to find corresponding location of target image by finding the max value of target image attention value. Related experiments show the effectiveness of the proposed method.

**Strengths:**

1.The idea of leveraging semantic knowledge of text-to-image diffusion model sounds interesting. This paper builds a new type of unsupervised semantic correspondence method based on text-to-image diffusion model.

2.The designed method is effective and achieve promising correspondence results on common datasets.

3.The design choices are reasonable, e.g., random crop for preventing overfitting when finding the text prompts.

**Weaknesses:**

1.The usage of text-to-image (t2i) diffusion model is not well-motivated. The proposed method mainly leverages the cross-modal ability of t2i diffusion model but underuses the generative ability of t2i diffusion model. The role of t2i diffusion model in proposed method can be replaced by other cross-modal pretraining model, e.g., CLIP, BLIP, GLIP. Please claim the unique significance for adopting t2i diffusion model rather than other cross-modal pretraining model.

2.The proposed method has to execute dozens of denoising steps for each query location. As diffusion model is notorious for its inference speed and cost, the total overhead is expensive. Are there any mitigation measures? For example, grouping the query locations or modifying the diffusion model architecture?

**Questions:**

Please see “Weakness” part.

**Limitations:**

The authors objectively discuss the limitations of the proposed method, e.g., the slow inference time caused by the introduction of diffusion model. This paper doesn’t have negative societal impacts.

---

> ### Author Rebuttal · Authors · 2023-08-10
>
> We thank the reviewer for the constructive feedback and for recognizing that our approach is effective and our results on unsupervised semantic correspondences are promising. We address the reviewer's comments below:
>
> ### **W-1: Missing motivation of using text-to-image (t2i) diffusion models and comparison with CLIP, BLIP, or GLIP networks.**
>
> Our work is motivated by the observation that text-to-image diffusion models are capable of generating high-quality images conditioned on textual prompts, indicating that these models understand the semantics of the objects being generated. This is owing to the specialized architecture used in Stable Diffusion which produces attention maps using cross-attention between textual and image features.
>
> It is indeed possible to leverage cross-modal models like BLIP within our proposed framework, although suboptimal. Taking into account the reviewer's suggestion, we use BLIP to get semantic correspondence between a pair of images by capturing the output from the cross-attention layers between the text embeddings and the image features. This is done similarly to our method across multiple layers in the model and aggregated. We get significantly worse performance on PF-Willow (PCK 13.1% for PCK @0.05 and 31.7% for PCK @0.1 ) in comparison to our method (PCK of 53.0% for PCK @0.05 and 84.3% for PCK @0.1). This underperformance may be attributable to factors such as the dataset size the model was trained on, the number of parameters in the model itself, or the model outputting text tokens as opposed to denoising the image. We thank the reviewer for this suggestion and will include these results in the revision.
>
> ### **W-2: Can grouping the query location reduce the computation overhead of SD inference?**
>
> This is an interesting observation. This would indeed reduce the computation overhead significantly. We leave this research to be explored in the future.

---

> > ### Comment · Reviewer_Vgvc · 2023-08-21
> >
> > Having read the rebuttal and the other reviews I decide to keep my initial rating. I hope the authors would address the time cost carefully.

---

> ### Author Response · Authors · 2023-08-21
>
> Thank you Reviewer Vgvc for your acknowledgement!
>
> We will for sure address the timing concerns in our future research. Preliminarily, we have found that grouping can indeed enhance speed significantly -- a group of correspondences and a single correspondences require similar runtime.
>
> Thanks,
> Authors

---

### Official Review · Reviewer_pziK · 2023-07-10

**Soundness:** 3 good
**Presentation:** 3 good
**Contribution:** 3 good
**Rating:** 5
**Confidence:** 4

**Summary:**

This paper explores unsupervised semantic correspondence tasks with stable diffusion model. Specifically, the authors proposed to first optimize the prompt embeddings of stable diffusion model, to maximize attention on the region of interest, then the optimized prompts are used for semantic correspondence.
Experimental results on PF-Willow, CUB-200 and SPair-71k datasets show that proposed method significantly outperforms weakly / unsupervised methods.



**Strengths:**

- This paper shows that one could explore diffusion models for semantic correspondence tasks.


- the writing is clear and easy to follow.

- The experiments show that the proposed method could even outperform supervised method on PF-Willow.


**Weaknesses:**



1. How important is the optimization for prompts? Could the authors show results if the prompts are a vanilla sentence, e.g. "an image of a cat"?
2. Concurrent works: there are multiple papers presenting correspondence ability of diffusion models:

[1*] Luo, Grace, et al. "Diffusion Hyperfeatures: Searching Through Time and Space for Semantic Correspondence." arXiv preprint arXiv:2305.14334 (2023).

[2*] Tang, Luming, et al. "Emergent Correspondence from Image Diffusion." arXiv preprint arXiv:2306.03881 (2023).

[3*] Zhang, Junyi, et al. "A Tale of Two Features: Stable Diffusion Complements DINO for Zero-Shot Semantic Correspondence." arXiv preprint arXiv:2305.15347 (2023).

Could the authors explain the differences if possible?



**Questions:**

please refer them to the weakness session

**Limitations:**

yes

---

> ### Author Rebuttal · Authors · 2023-08-10
>
> We thank the reviewer for the positive feedback and for recognizing that our approach achieves state-of-the-art results. We address the reviewer's concerns below:
>
> ### ***W-1: Importance of prompt-optimization for the correspondence task***
>
> Optimization of the prompts is critical. Without it, one cannot truly “localize”. As an example, we are attaching the attention map corresponding to the prompt “An image of a cat”. For the token “cat” the entire body of both cats is attended to rather than a single cat or parts of the cat. See respectively **Figures 1 and 4** in the provided rebuttal PDF above.
>
> ### ***W-2: Comparison with papers submitted to Arxiv after Neurips deadline***
>
> We note that the papers mentioned by the reviewer appeared on Arxiv after the NeurIPS submission deadline, and we believe that, per the NeurIPS guidelines, they should be counted as contemporary work and should not affect the evaluation of our paper.
>
> Nonetheless, in the revision, we plan to mention them as they are quite relevant, and in fact, highlight the core difference that our method has compared to them -- they all look into how to use the deep features within the Stable Diffusion Network effectively, similarly to how VGG19 features are widely used for various tasks.  Our method, on the other hand, looks into how we can alter the attention maps within Stable Diffusion to our will, in other words taking into account how these features are supposed to be used within Stable Diffusion. To do so we optimize embeddings. And by doing so we show that one can perform alternative tasks than simple image creation such as the semantic correspondence task we demonstrated. However, this is not the end of what our framework can do as *reviewer 5igN* pointed out. For example, an immediate straightforward extension of our method could be to learn an embedding for a part of the cat, e.g., paw, using multiple images and not just one. We note that our current work was a first investigation about whether this was at all possible, and that whether one solution is more general and useful than the other, is a question for future works.
>
> In more detail, Diffusion Hyperfeatures [1*] consolidates multi-scale and multi-timestep feature maps from Stable Diffusion into per-pixel feature descriptors with a lightweight aggregation network. A Tale of Two Features [3*] introduces a fusion approach that capitalizes on the distinct properties of Stable Diffusion (SD) features and DINOv2 by extracting per-pixel features from each. Emergent Correspondence from Image Diffusion [2*] extracts per pixel features from Stable Diffusion.
>
> [1*] Luo, Grace, et al. "Diffusion Hyperfeatures: Searching Through Time and Space for Semantic Correspondence." arXiv preprint arXiv:2305.14334 (2023).
>
> [2*] Tang, Luming, et al. "Emergent Correspondence from Image Diffusion." arXiv preprint arXiv:2306.03881 (2023).
>
> [3*] Zhang, Junyi, et al. "A Tale of Two Features: Stable Diffusion Complements DINO for Zero-Shot Semantic Correspondence." arXiv preprint arXiv:2305.15347 (2023).

---

### Official Review · Reviewer_5igN · 2023-07-30

**Soundness:** 3 good
**Presentation:** 3 good
**Contribution:** 2 fair
**Rating:** 5
**Confidence:** 4

**Summary:**

The paper proposes an approach for unsupervised semantic correspondence by employing a pre-trained Stable Diffusion generator. The basic idea is that since text-to-image diffusion models are capable of generating realistic images, they must understand the semantics of the objects they generate and thus be capable of finding semantic similarities in images.

Technically, the approach feeds a real image (or, better, its latent vector) into a diffusion pipeline, recover attention matrices between pixel values and the textual prompt, and optimizes the textual prompt embedding so that the cross-attention focuses on a query pixel location. Once optimization is over, the same text embedding can be employed to find semantic correspondences in a target image. Cross-attention matrices are averaged across channels and a subset of U-Net layers, using bilinear interpolation. Further, the target attention map is simulated with a Gaussian distribution centered on the query location, and a L2 loss is employed for optimization.

Authors also propose two regularization strategies to prevent over-fitting during optimization, i.e. averaging across crops of the query image, and averaging across multiple rounds of optimizations/initialization choices.

Experimental results are conducted over three standard benchmarks for the task, i.e. SPair-71k, PF-Willow and CUB-200. The proposed approach achieves superior results when compared to unsupervised or weakly-supervised approaches, and is also competitive when compared to models employing strong supervision.

**Strengths:**

- The main idea of the paper (extracting/exploiting the semantics embedded in a pre-trained a SD network) is novel and interesting. It is in-line with some recent pre-prints working on related tasks (e.g. image segmentation), but to my knowledge the proposed approach is novel wrt published papers.
- The technique for extracting semantics is simple and effective, and also are the two regularization techniques.
- Experimental results are solid and confirm the appropriateness of the approach.
- The paper is well written, clear and a pleasure to read. The supplementary material is also interesting and comprehensive.

**Weaknesses:**

- While the idea behind the paper is very general and could have been applied to a variety of tasks involving semantics (e.g. image segmentation, panoptic segmentation, classification), the paper only focuses on finding semantic correspondences. This limits the overall impact of the work to a single task of Computer Vision. I wonder if authors have tried to extend to other tasks, and how difficult that would be.

**Questions:**

See above.

**Limitations:**

The authors have successfully discussed the limitations of the work in the appropriate section.

---

> ### Author Rebuttal · Authors · 2023-08-10
>
> We thank the reviewer for the constructive feedback and for recognizing that our proposed approach is novel, effective and our experimental results are solid. We address the reviewer's comments below:
>
> ### ***Discussion on the extension of the approach to tasks such as semantic segmentation, classification etc.***
>
> We thank the reviewer for suggesting an extension to this work. Indeed image segmentation tasks could possibly be accomplished by optimizing an embedding to attend to a region as opposed to a single point.  We plan to continue exploring potential extensions -- as the reviewer pointed out, our framework is indeed very general after all, and can be seen as a direction forward for extracting learned knowledge from these large generative models.

---

> > ### Comment · Reviewer_5igN · 2023-08-18
> >
> > I acknowledge that I have read the response, and I intend to keep my original rating.

---

> ### Author Response · Authors · 2023-08-21
>
> Thank you Reviewer 5igN for your acknowledgement!
>
> We would also like to share that, as you guessed, preliminary results show that semantic segmentation is also something that is quite possible. We are also seeing positive signs for other applications as well. While we cannot be specific about them at the moment due to internal reasons, we wanted to share our enthusiasm with you.
>
> Thanks,
> Authors

---

### Author Rebuttal · Authors · 2023-08-10

We thank the reviewers for their constructive feedback. We are glad that the reviewers acknowledge that our method is novel and provide new insights, with solid experimental results, and clear exposition. We provide detailed responses in the individual rebuttals and provide the figures and the tables in the attached PDF.

---

### Decision · Program_Chairs · 2023-09-21

**Decision:**

Accept (poster)

**Comment:**

This dissertation deals with unsupervised semantic correspondences between a pair of images. They propose to exploit the semantic knowledge within fashionable text-image diffusion model to accomplish this task. By optimizing the text prompt embeddings of the stable diffusion model, such a correspondence can be read out via attention maps in the deep network. The resulting optimized prompts are used for semantic correspondence.  Related experiments on PF-Willow, CUB-200 and SPair-71k datasets clearly demonstrate the effectiveness of the proposed method. All reviewers emphasized the originality of the proposed work, but also asked for explanations and clarifications.
After a period of rebuttal and discussion, all were satisfied with the answers. Although it is a pity that the article focuses solely on semantic correspondences, this paper makes a solid contribution that should be accepted.